# The Response of Triple-Negative Breast Cancer to Neoadjuvant Chemotherapy and the Epithelial–Mesenchymal Transition

**DOI:** 10.3390/ijms24076422

**Published:** 2023-03-29

**Authors:** Stefano Zapperi, Caterina A. M. La Porta

**Affiliations:** 1Center for Complexity and Biosystems, Department of Physics, University of Milan, Via Celoria 16, 20133 Milano, Italy; 2CNR—Consiglio Nazionale delle Ricerche, Istituto di Chimica della Materia Condensata e di Tecnologie per l’Energia, Via R. Cozzi 53, 20125 Milano, Italy; 3Center for Complexity and Biosystems, Department of Environmental Science and Policy, University of Milan, Via Celoria 10, 20133 Milano, Italy; 4CNR—Consiglio Nazionale delle Ricerche, Istituto di Biofisica, Via Celoria 10, 20133 Milano, Italy

**Keywords:** triple-negative breast cancer, drug response, epithelial–mesenchymal transition

## Abstract

It would be highly desirable to find prognostic and predictive markers for triple-negative breast cancer (TNBC), a strongly heterogeneous and invasive breast cancer subtype often characterized by a high recurrence rate and a poor outcome. Here, we investigated the prognostic and predictive capabilities of ARIADNE, a recently developed transcriptomic test focusing on the epithelial–mesenchymal transition. We first compared the stratification of TNBC patients obtained by ARIADNE with that based on other common pathological indicators, such as grade, stage and nodal status, and found that ARIADNE was more effective than the other methods in dividing patients into groups with different disease-free survival statistics. Next, we considered the response to neoadjuvant chemotherapy and found that the classification provided by ARIADNE led to statistically significant differences in the rates of pathological complete response within the groups.

## 1. Introduction

Triple-negative breast cancer (TNBC) is a subtype of breast cancer characterized by the lack of expression of estrogen receptors (ERs), progesterone receptors (PRs) and human epidermal growth factor receptor 2 (HER-2). According to epidemiological studies, TNBC is most prevalent in premenopausal young women that are less than 40 years old [1]. An important feature of TNBC is the heterogeneity of the tumor across different patients, which is then reflected in great variability in clinical outcomes [2,3]. TNBC is, in general, highly invasive and approximately 40% of TNBC patients will have distant metastases, leading, in that case, to a median survival time of only around 13 months and a high recurrence rate after surgery of about 25% [4,5,6].

Due to its special molecular phenotype, TNBC is not sensitive to endocrine therapy or molecular targeted therapy. TNBC is, however, considered to be chemotherapy-sensitive, so that neoadjuvant chemotherapy, followed by surgery, remains the standard of care for newly diagnosed early TNBC. Neoadjuvant chemotherapy typically includes a combination of taxanes (mitotic inhibitors) and anthracyclines (DNA intercalators), and it is often administered intensively, with sequential regimens of anthracyclines and taxanes [7]. Furthermore, platinum-based chemotherapy is sometimes also used in combination with these treatments [8]. Pathological complete response (pCR) is verified after surgery and defined as the absence of disease in the breast or lymph nodes. Clinical studies show that pCR in TNBC is associated with a better clinical outcome, with significant increases in the disease-free survival times [9]. In more detail, the pCR rates for anthracyclines alone range from 14% to 47% [10], while for sequential anthracycline and taxane regimens the range is between 17% and 39% [11,12]. Another study reported pCR rates that reached up to 57% for TNBC managed with neoadjuvant anthracyclines, cyclophosphamide and taxanes [13]. Finally, clinical studies of neoadjuvant chemotherapy supplemented by carboplatin generally report an increased pCR [14,15]. Given that pCR provides a relatively good early indication of the long-term disease outcome, clinical trials have attempted to define the combination of systemic agents that could maximize pCR [7].

In recent years, a great effort in the scientific community has been devoted to the identification of specific targets for TNBC, in particular for the metastatic tumorm which continues to lead to a poor prognosis. New drugs for the treatment of metastatic TNBC are currently under clinical trial, such as poly (ADP-ribose) polymerase (PARP) inhibitors and immunotherapy agents [16,17,18]. One of the key issues, however, remains the identification of subpopulations of TNBC patients for which a specific treatment is most effective [19,20]. Genomic tests represent a potential solution to this issue since they can potentially stratify tumors according to their phenotype, which is summarized by the transcriptome. Tests based on gene expression data are widely available for the luminal A breast cancer subtype. These tests are based on the expression level of empirically selected gene panels [21,22] or on gene sets obtained from the machine learning classification of the transcriptomes of a patient cohort [23,24]. Machine learning-based tests suffer, however, from overfitting because the algorithm tries to classify a high-dimensional object (i.e., the transcriptome) using a relatively small training set (typically a cohort of hundreds or at most a few thousand patients) [25,26]. We have discussed these issues in a recent paper where we suggested that effective strategies should try to combine novel algorithms with biological insight, while a straightforward application of machine learning is limited by the relatively scarce availability of transcriptomic data for breast cancer patients [27].

Gene expression tests specifically designed for TNBC have emerged in the last decade [28,29,30]. A data-driven classification scheme for TNBC was proposed and then refined [28,31] by Lehmann et al. who empirically clustered gene expression data into four subtypes [28] and then refined the classification into six subtypes [31]: basal-like 1, basal-like 2, immunomodulatory, mesenchymal, mesenchymal stem-like and luminal androgen receptor (LAR) subtypes. Each subtype is distinguished by its gene expression patterns and biological features. The basal-like subtypes exhibit high expression of basal markers and are considered more aggressive, while the immunomodulatory and LAR subtypes are believed to be less aggressive and associated with better outcomes. The mesenchymal and mesenchymal stem-like subtypes are characterized by increased invasiveness and metastatic potential. There is evidence that these six TNBC subtypes display differential responses to treatment [31], but no statistically significant differences in relapse-free survival has been found [31].

We recently followed an alternative strategy to stratify TNBC patients that is based on the identification of hybrid epithelial/mesenchymal phenotypes from gene expression data [32]. The epithelial–mesenchymal transition (EMT) describes the transformation of polarized epithelial (E) cells into mesenchymal (M) cells where cell polarity is lost and adhesion molecules are down-regulated. Since M cells tend to be more motile than E cells, it is generally assumed that EMT would promote metastasis [33,34,35,36]. This picture was then refined by the identification of hybrid E/M states [37], where cells display markers that are characteristic of both E and M cells [38,39,40]. Hybrid phenotypes combine adhesion and invasive capabilities [41,42] and are associated with tumor aggressiveness [37,43,44,45,46]. Our method, ARIADNE, is able to efficiently classify cell phenotypes by mapping gene expression data into the state of a Boolean network model of the EMT pathway [47]. In particular, we showed that the score obtained with ARIADNE is able to classify TNBC patients into groups with different disease-free survival statistics [32]. We showed that ARIADNE correlates with other EMT scores [46], but it is more specific to identifying hybrid phenotypes, which is essential to stratify patients [32]. In a recent paper [48], we also showed that ARIADNE is able to identify high-risk patients that were not captured by recent classification schemes based on the tumor immune microenvironment [49].

In this paper, we extended previous studies [32,48] and compared the prognostic performance of ARIADNE with that of other standard pathological classification methods, such as the histological tumor grade (G), the tumor stage (T) and the nodal status (N). These three critical factors are commonly used to assess the extent and severity of breast cancer. A tumor grade refers to the histological assessment of the cancer cells’ appearance and differentiation under a microscope. The grade is determined based on the degree of abnormality and how closely the cancer cells resemble the surrounding normal tissue. Tumor grade is reported on a scale of 1 to 3, with grade G1 indicating well-differentiated cancer cells and grade G3 indicating poorly differentiated or undifferentiated cancer cells. A tumor stage refers to the extent of the primary breast cancer and is determined by the size of the tumor, its invasiveness and whether it has spread to nearby tissue or organs. The tumor stage is classified on a scale of T0 to T4, with stage T0 indicating non-invasive breast cancer and stage T4 indicating metastatic breast cancer that has spread to other parts of the body. Finally, nodal status refers to the involvement of the lymph nodes near the breast. The lymph nodes act as filters for the lymphatic fluid draining from the breast, and cancer cells can spread to these nodes through the lymphatic system. The nodal status is determined by assessing the number of lymph nodes involved and the extent of cancer cell infiltration within the lymph nodes. The nodal status is classified on a scale of 0 to 3, with N0 indicating no lymph node involvement and N3 indicating extensive involvement of lymph nodes. Together, G, T and N are used to stage breast cancer and determine its severity and prognosis. The combination of these factors assist clinicians in developing individualized treatment plans, with the goal of achieving the best possible outcomes. In general, early-stage breast cancer (T1 or T2) with no nodal involvement (N0) and a low tumor grade (G1 or G2) has a better prognosis than later-stage cancer with higher tumor grades and nodal involvement.

Next, we investigated the predictive ability of ARIADNE for the response of TNBC patients to neoadjuvant chemotherapy. This is a crucially important issue to determine the most appropriate treatment plan, improve patient outcomes and reduce the risk of recurrence. In particular, it would be important to identify patients who are unlikely to respond to chemotherapy, enabling clinicians to explore alternative treatment options, such as targeted therapies, immunotherapy or clinical trials. Patients who display pCR may require less adjuvant therapy, while those with residual disease may require more aggressive adjuvant treatments to reduce the risk of recurrence.

## 2. Results

### 2.1. The Prognostic Value of ARIADNE for TNBC in Comparison with Grade, Stage and Nodal Status

As discussed in the introduction, widely employed pathological classiffiers for TNBC are the grade (G), the stage (T) and the nodal status (N). The tumor grade is obtained from histology and is associated with the typical growth patterns of the tumor. A low grade (G1) indicates that the tumor is slow growing, while intermediate (G2) and high grade (G3) tumors are more and more likely to grow. The T stage is used instead to classify the tumor size and whether it has spread into nearby tissue, ranging from T1 to T4 in increasing order of severity. Finally, the nodal status (N) describes whether the tumor has spread to the lymph nodes.

The gene expression data reported in GSE31595 include the metadata of disease-free survival times and the classification of the tumor according to the G, T and N scores for 383 TNBC patients. We, therefore, investigated how this classification was related to ARIADNE. In Figure 1a, we report the Kaplan–Meyer survival curves for TNBC patients stratified according to their grade. As expected, low grade tumors had slightly longer disease-free survival, although the results were not statistically significant. We then stratified the two grade classes (G1 vs G2/G3) with ARIADNE and found significant differences in disease-free survival between ARIADNE-low and ARIADNE-high patients for both the G1 and G2/G3 groups (see Figure 1b,c, respectively).

In Figure 1d, we consider the T stage showing that it also stratified patients into two classes with small variations in disease-free survival depending on the stage. In addition, in this case, we stratified the two groups with ARIADNE that were previously stratified according to their T-stage. For both stage T0 and higher, ARIADNE was able to stratify the groups with significant differences in disease-free survival times (see Figure 1e,f). Finally, we repeated the same procedure considering the nodal status N. As shown in Figure 1g,h, the outcome was similar to the previous cases: when patients were stratified according to their nodal status, the difference in disease-free survival was modest, while when each group was stratified further with ARIADNE, we observed considerable differences in disease-free survival.

### 2.2. Evaluation of the Predictive Value of ARIADNE for the Response to Neoadjuvant Chemotherapy

Having established that ARIADNE is able to stratify patients into groups with different outcomes in disease-free survival, we investigated if the ARIADNE score was related to the response of the patient to neoadjuvant chemotherapy. Previous clinical studies showed that pCR is associated with better prognosis with improved disease-free survival [9]. Here we considered gene expression data reported in GSE25066, which includes information about survival and response for 183 TNBC patients treated with anthracycline/taxane in the metadata. Figure 2a summarizes the results of the pCR fraction of each ARIADNE group, compared with the baseline provided by the entire population. The results showed that patients in the ARIADNE-low subgroup had a smaller rate of pCR than the baseline (25% against 35%, but statistically not significant). The outcome in terms of disease-free survival was still slightly better for the ARIADNE-low subgroup with respect to the ARIADNE-high subgroup, although the result was not statistically significant. This was probably due to the limited sample size (see Figure 2b). To better understand these results, we performed additional investigations. In Figure 3a, we reported the survival curves of the patients divided into groups according to their response to the treatment. The results showed that, as expected, patients with pCR had considerably longer survival times than patients with residual disease (RD). We then further stratified the two groups with ARIADNE and showed that 100% of the patients with pCR and in the ARIADNE-low group survived for the entire time frame, see Figure 3b. On the other hand, patients in the ARIADNE-high group had reduced survival. Figure 3c reports a similar analysis for the RD group stratified according to ARIADNE, showing again that patients in the ARIADNE-low group had increased survival, although the difference was lower.

To increase the statistics, we also considered data from GSE106977, which reported the response of 119 TNBC patients treated with anthracycline/taxane alone (88 patients) or in combination with carboplatin (31 patients). We classified these patients according to ARIADNE and summarized the results, see Figure 4a–c. Again, the rate of pCR for the ARIADNE-low group was smaller than the baseline value: a result that held for both types of treatment. We finally combined the results from GSE25066 and GSE106977 and considered all the TNBC patients treated with anthracycline/taxane alone (a total of 271 patients). We found that for patients belonging to the ARIADNE-low group the total rate of pCR was 23%, while for those belonging to ARIADNE-medium or ARIADNE-high groups the pCR rate was 44% (see Figure 4d). A Fisher exact test established that the result was statistically significant with p=0.005.

## 3. Discussion

Triple-negative breast cancer (TNBC) is a subtype of breast cancer characterized by the absence of hormone receptors (estrogen receptors (ERs) and progesterone receptors (PRs)) and human epidermal growth factor receptor 2 (HER2). TNBC accounts for 15–20% of all breast cancer cases and is more frequently diagnosed in premenopausal young women and those carrying BRCA1 gene mutations [1]. Due to the lack of therapeutic targets, TNBC is challenging to treat, and chemotherapy is the standard of care. However, not all TNBC cases respond to chemotherapy, and patients may experience adverse side effects. TNBC patients have a poorer prognosis compared to other breast cancer subtypes, with a higher risk of distant recurrence and lower overall survival rates.

TNBC is a heterogeneous disease with different subtypes exhibiting varying degrees of aggressiveness and responsiveness to treatment [2,3]. To improve outcomes for TNBC patients, various approaches have been explored, including targeted therapies, immunotherapy and identification of biomarkers [28,29,30]. Immunotherapy, which enhances the immune system’s ability to recognize and attack cancer cells, is a promising strategy for TNBC treatment. Various immunotherapeutic agents, such as checkpoint inhibitors, monoclonal antibodies and chimeric antigen receptor (CAR) T cells, are currently undergoing clinical trials for TNBC. TNBC’s heterogeneity has led investigations to its genetic and molecular characteristics, which have led to the identification of potential biomarkers. For instance, TNBC patients with BRCA1 gene mutations may have a better response to platinum-based chemotherapy [50].

In this paper, we evaluated the prognostic and predictive capabilities of ARIADNE, a test for TNBC that is based on the identification of hybrid E/M phenotypes from gene expression data. The EMT is a physiological process involved in embryonic development and wound healing. In cancer, EMT is a critical process that contributes to tumor progression, invasion and metastasis. EMT involves the transformation of cancer cells from an epithelial phenotype to a mesenchymal phenotype, characterized by increased cell motility, invasiveness and resistance to apoptosis. EMT is triggered by various signaling pathways, including the TGF-β pathway, the Wnt/β-catenin pathway and the Notch signaling pathway, among others. These pathways activate transcription factors, such as Snail, Slug and Twist, which downregulate the expression of epithelial markers and upregulate the expression of mesenchymal markers. Hybrid E/M cancer cells exhibit enhanced migratory and invasive abilities and increased resistance to chemotherapy and targeted therapies.

ARIADNE is based on a Boolean network model of the EMT, which provides a landscape of possible cell phenotypes [47]. Gene expression data from individual patients can be projected on the phenotypic landscape and scored according to the prevalence of the hybrid E/M phenotype [32]. In this way, TNBC patients can be classified in different score groups, which are associated with the risk of aggressiveness of the tumor. In previous studies, ARIADNE was shown to be able to evaluate the aggressiveness of TNBC and provide prognostic information about the long-term critical outcome of patients in terms of disease-free survival [32,48], but a comparison with other standard pathological classifiers or a study of drug responses was not previously attempted. Here we have filled this gap by comparing the prognostic ability of ARIADNE with standard pathological classifiers that are currently employed in clinical practice, such as the grade, stage and nodal status, and studied the ability of ARIADNE to predict the response to neoadjuvant chemotherapy.

Our results clearly showed that the prognostic value of ARIADNE is stronger than those of any of the standard classifiers. In particular, ARIADNE is able to further stratify patients within all the groups established on the basis of grade, stage or nodal status. Furthermore, the difference in survival times was generally larger between patients in the ARIADNE-low/high groups than the one between groups established using the other classifiers.

We next evaluated the ability of ARIADNE in predicting the response to neoadjuvant chemotherapy in TNBC patients. We did this by combining the results of two different studies that reported the response to anthracycline/taxane, mostly alone, and combined with carboplatin for a small number of patients. We found that patients classified in the low risk group by ARIADNE displayed a lower pCR rate than patients in the other ARIADNE groups. This result was consistently found in both of the studies considered. While the number of patients in each separate study was too small to yield statistical significance, when we combined the results of the two studies we found that patients classified in the ARIADNE-low group were significantly less likely to achieve pCR than the other patients (23% versus 44%, with p=0.005, according to Fisher’s exact test). This result appeared to be counter-intuitive given that patients classified as low risk by ARIADNE displayed statistically better prognosis in terms of disease-free survival. We clarified this apparent contradiction by computing the survival curves of ARIADNE groups within responding and non-responding patients. For each of the patient groups (responding and non-responding), ARIADNE-low patients had generally better prognosis. In particular, for patients achieving pCR, all patients classified as low risk by ARIADNE survived for the whole time frame of the study.

We believe that patients selected by ARIADNE as low risk respond less to neoadjuvant chemotherapy because the biological features highlighted by ARIADNE are not directly related to the mechanism of action of the drugs. Anthracycline and taxane target proliferating cancer cells, while ARIADNE selects hybrid E/M phenotypes that are more prone to lead to metastasis. The prognostic abilities of ARIADNE should then be related to the fact that survival is mostly dictated by the occurrence of metastasis. It is interesting to note that when pCR was achieved ARIADNE could still predict disease outcome in terms of long-term survival. It would be extremely interesting to test the predictive capabilities of ARIADNE for new drugs recently developed to target metastasis in TNBC.

## 4. Materials and Methods

### 4.1. Gene Expression Data

We considered three gene expression series of TNBC patients present in the Gene Expressiom Omnibus (GEO) database: GSE31519, GSE25066 and GSE106977. GSE31519 contains gene expression data for 579 TNBC patients, with survival information for 383 of them. The metadata also include the grade (either G1 or G2/G3), the T stage (either T1 or T2,T3,T4) and the nodal status (either N0 or N1). These data were already analyzed with ARIADNE in [32] without considering the GTN information. In GSE25066, the authors reported gene expression data from 189 TNBC patients and provided information about the responses to anthracycline/taxane for 183 of those patients [51]. The metadata also reported survival information. In GSE10677, the authors performed a retrospective analysis of paraffined pretreatment tumor biopsies from 119 TNBC patients (88 patients treated with neoadjuvant anthracyclines and/or taxanes and 31 with anthracyclines and/or taxanes plus carboplatin) [52]. A summary of the data analyzed in the present paper is reported in Table 1.

### 4.2. Data Normalization and Computation of ARIADNE Scores

We normalized data from GSE25066 by following the procedure adopted by [53] for GSE31519:1.log2 transformation of MAS5 values;2.Median centering of arrays;3.Magnitude normalization of arrays.

In the above, magnitude normalization must be understood as setting the sum of squares of all samples to one. This is because ARIADNE was trained on GSE31519 [32] and, as we already performed this in [48], we did not retrain ARIADNE but reused the parameters obtained in [32] to compute the score of a new dataset. To this end, it was crucial to use the same normalization technique as in the training data. We checked that the distribution of normalized expressions for the genes used in the score computation in the training data of ARIADNE (i.e., GSE31519) were in good agreement with the data that was newly analyzed in the present paper (i.e., GSE25066). We then computed the ARIADNE score as explained in [32] for the samples in GSE25066. After computing the raw ARIADNE score, which was an integer value, we defined ARIADNE groups by sorting and splitting the dataset into three groups (high, medium and low) based on quantile discretization (with intervals [0,0.15], (0.15,0.85] and (0.85,1], respectively) as in [32]. In the case of GSE106977, gene expression data were not provided as MAS5 values as in GSE31519 and GSE25066 but rather as RMA values. To avoid problems due to the different platforms employed, we recomputed the ARIADNE landscape for these samples following [32]. In addition, in this case, we computed the raw ARIADNE scores and defined ARIADNE groups by sorting and splitting the dataset into three groups, as discussed above.

### 4.3. Computation of Survival Curves

We used the lifelines Python package to compute survival curves using the Kaplan–Meyer approach.

### 4.4. Statistical Analysis

Statistical analysis for the survival curves was performed using the logrank test as implemented in the lifelines Python package. Differences among groups for pathological complete response was evaluated using the Fisher exact test, as implemented in the SciPy package.

## 5. Conclusions

In this work, we studied the performance of the ARIADNE test for TNBC by evaluating the response to neoadjuvant chemotherapy using publicly available data. We found that patients with a low ARIADNE score were less likely to display pCR, but patients with pCR had increased disease-free survival. Hence, ARIADNE could be useful in predicting the long-term survival of TNBC patients with pCR undertaking neoadjuvant chemotherapy. We also compared ARIADNE with patient stratification based on standard pathological classifiers such as grade, nodal status and tumor stage and found that ARIADNE was more effective in predicting disease-free survival. Further studies are needed to confirm our results with improved statistical significance.

## Figures and Tables

**Figure 1 ijms-24-06422-f001:**
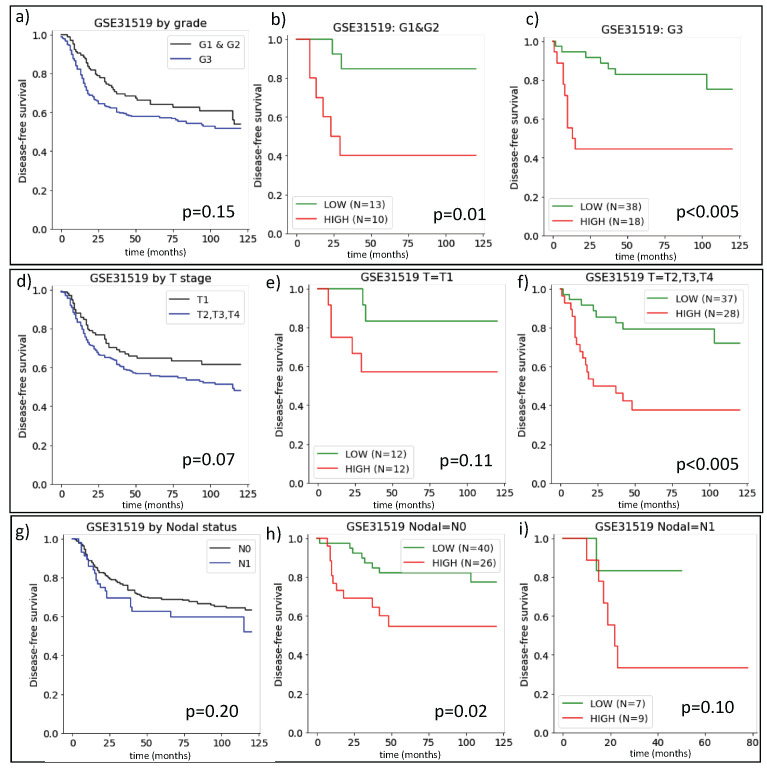
Stratification of TNBC patients according to standard pathologial classifications in grades and stages compared with EMT-based ARIADNE classification: Kaplan–Meyer plots for TNBC patients stratified according to (**a**–**c**) tumor grade G, (**d**–**f**) tumor stage T and (**g**–**i**) nodal (N) status. We report the survival curves first stratified by G, T and N (**a**,**d**,**g**) and then the stratification of individual pathological classifications (i.e., by G, T,N) by ARIADNE. The results showed that ARIADNE was able to better stratify patients, even within individual categories.

**Figure 2 ijms-24-06422-f002:**
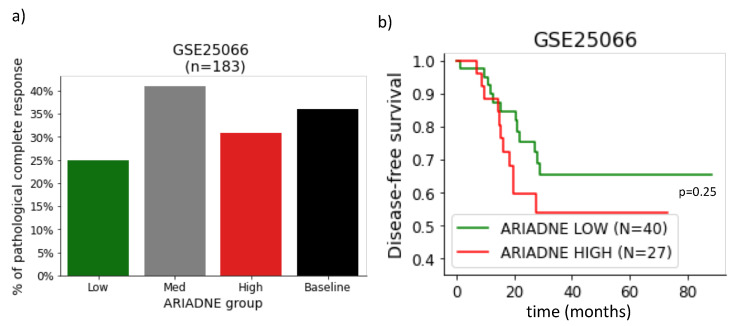
Response to neoadjuvant chemotherapy for different ARIADNE groups for dataset GSE25066: (**a**) percentage of patients with pathological complete response for the three different ARIADNE groups compared with the baseline provided by the entire population, the difference between the ARIADNE-low group and the rest of the patients was not statistically significant (p=0.07); (**b**) stratification of the same population according to ARIADNE.

**Figure 3 ijms-24-06422-f003:**
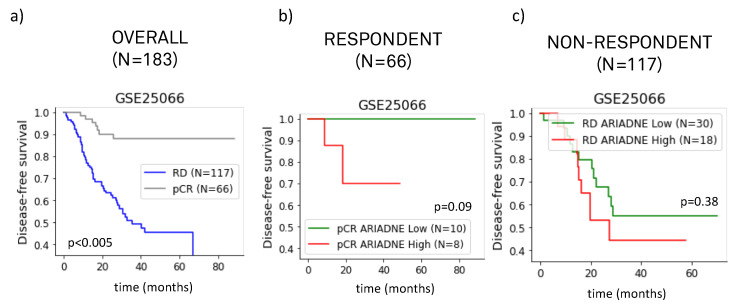
Survival curves of TNBC patients depending on their response to neoadjuvant chemotherapy for dataset GSE25066: (**a**) patients that responded to chemotherapy (pCR) had longer survival than those with residual disease (RD); (**b**) survival curves showing that among the patients with pathological complete response all those belonging to the low risk group according to ARIADNE survived for the considered time period, while survival was reduced for those belonging to the high-risk group; (**c**) the difference in survival among ARIADNE groups was smaller for patients that did not respond.

**Figure 4 ijms-24-06422-f004:**
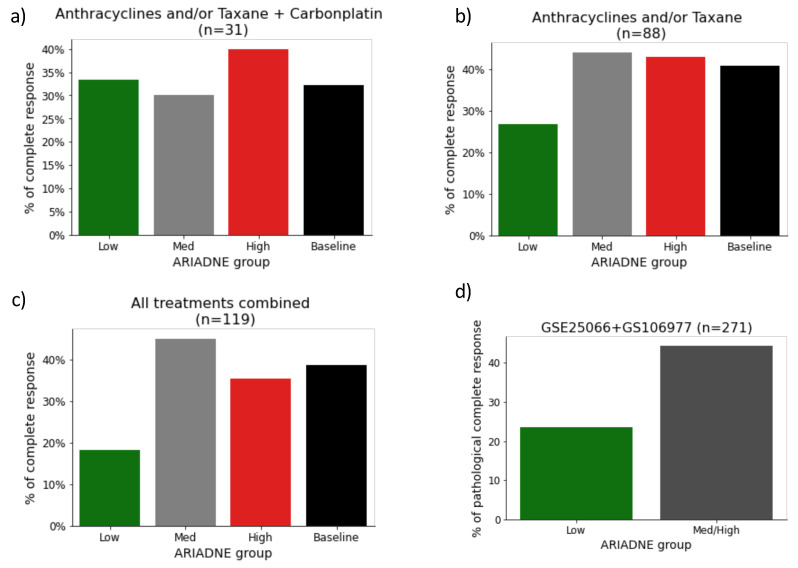
Response to neoadjuvant chemotherapy with and without carboplatin for different ARIADNE groups for dataset GSE106977: Percentage of patients with pathological complete response for the three different ARIADNE groups compared with the baseline provided by the entire population for (**a**) all treatment combined, (**b**) only treated with anthracycline and/or taxane, (**c**) treated with anthracycline and/or taxane plus carboplatin; (**d**) response to neoadjuvant chemotherapy, combining with patients from GSE106977 (excluding those treated with carboplatin) with those from GSE25066. A Fisher exact test on the two groups yielded p=0.005.

**Table 1 ijms-24-06422-t001:** Summary of the gene expression data analyzed in the present paper.

Dataset	No. Patients	Treatment	Analysis Performed
GSE31519	383	Not available.	Statification, according to GNT and survival.
GSE25066	183	Anthracycline/taxane.	Drug response and survival.
GSE10677	119	Anthracycline/taxane and carboplatin.	Drug response.

## Data Availability

The datasets analyzed for this study are available in the Gene Expression Omnibus (GEO) database (https://www.ncbi.nlm.nih.gov/geo/) under accession numbers GSE31519, GSE25066 and GSE106977 (last accessed 22 September 2022).

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
