# Peer review of "The Response of Triple-Negative Breast Cancer to Neoadjuvant Chemotherapy and the Epithelial–Mesenchymal Transition"

_ijms, 2023, doi:10.3390/ijms24076422_

Round 1

Reviewer 1 Report

minor issues

1.     Figure 1. Is a there a unit for the x-axis timeline?

2.     References should be checked carefully, and arranged according to the journal requirements.

Author Response

  1. Figure 1. Is a there a unit for the x-axis timeline?
    The units are "months". We have now replaced all the x-axis label to show this.
  2. References should be checked carefully, and arranged according to the journal requirements.
    We have now reformatted the references according to the house style.

Reviewer 2 Report

The authors use the ARIADNE algorithm published by Font-Clos et al., 2021. Font-Clos et al. established the ARIADNE algorithm to map gene expression data for triple-negative breast cancer patients and the possibility of stratifying patients according to their prognosis.

·      The authors used the ARIADNE algorithm and compared it with other methods but did not discuss the other methods. Please include which methods were used.

·      They should have explained the differences with Font-Clos et al., 2021; please include and compare the differences between the two articles.

·      The patient number has to be increased for the study

1)    Classification of triple-negative breast cancers through a Boolean network model of the epithelial-mesenchymal transition” by Font-Clos et al., 2021

Author Response

  • The authors used the ARIADNE algorithm and compared it with other methods but did not discuss the other methods. Please include which methods were used.

    The other methods are the histological tumor grade (G), the tumor stage (T) and the nodal status (N).  These are standard methods to classify TNBC. This is discussed on line 76 of the revised manuscript.
  • They should have explained the differences with Font-Clos et al., 2021; please include and compare the differences between the two articles.

    In Font-Clos et al 2021 we presented ARIADNE and showed that it could stratify TNBC into classes with different disease free survival statistics.
    The present paper extends the analysis to the following cases not considered in Fong-Clos et al 2021
    1) Stratification of patients according to G,T,N criteria and comparison with ARIADNE, including stratification of patients with ARIADNE for patients already stratified according to G,T,N criteria. (Fig. 1)
    2) Study of the chemotherapy response in groups stratified with ARIADNE (Fig. 2).
    3) Stratification with ARIADNE of groups stratified according to their response to chemotherapy (Fig. 3).
    4) Response to chemotherapy and carboplatin in groups stratified with ARIADNE (Fig. 4).

    Hence all the figures report original results on aspects not previously considered in Font-Clos et al 2021.  We mention this on line 193.
  • The patient number has to be increased for the study
    We analyzed all the samples that we could find publicly available that contained response to chemotherapy. Unfortunately, we could not access additional patient data at this stage.

Reviewer 3 Report

*The article presented here is very interesting, but in the section Gene expression data, of Materials and Method, a table could be included where the characteristics that have been studied in the research such as: Grade, T stage, nodal status, response to treatments, and how many patients are included in each of them, with respect to GSE31519, GSE25066 and GSE106977. This would help us to better understand what is done afterwards.

* A Figure 4a could be put up with the combination of the results from GSE25066 and GSE106977.  In this way, we could better understand what has been explained in the text.

Author Response

The article presented here is very interesting, but in the section Gene expression data, of Materials and Method, a table could be included where the characteristics that have been studied in the research such as: Grade, T stage, nodal status, response to treatments, and how many patients are included in each of them, with respect to GSE31519, GSE25066 and GSE106977. This would help us to better understand what is done afterwards.

We now provide a table summarizing this information (table 1).

  • A Figure 4a could be put up with the combination of the results from GSE25066 and GSE106977.  In this way, we could better understand what has been explained in the text.

We now include an additional panel to Fig. 4 (panel d) incorporating the results of GSE25066 and GSE106977.

Reviewer 4 Report

Conclusion:

The discussion and conclusion is together. I advice to make separate conclusion

References: 

Only 1 2022 reference has been cited.

Advised to cite 2023 references as well

Example (not necessary cite this -- Cite related relevant references):

Hayashi T, Kobayashi N, Ushida K, Asai N, Nakano S, Fujii K, Ando T, Utsumi T. Effect of eribulin on epithelial–mesenchymal transition plasticity in metastatic breast cancer: An exploratory, prospective study. Genes to Cells. 2023 Feb 27.

Ibrahim FM, Helal DS, Ali DA, Abd-Ellatif RN, Elkady AM, Sharshar R, Gharib F, Elnasr MA, El-Guindy DM. Prognostic role of annexin A2 and cancer-associated fibroblasts in advanced non-small cell lung cancer: Implication in epithelial-mesenchymal transition and gefitinib resistance. Pathology-Research and Practice. 2023 Jan 1;241:154293.

Author Response

The discussion and conclusion is together. I advice to make separate conclusion

We now include a conclusions section.

References:  Only 1 2022 reference has been cited. Advised to cite 2023 references as well

We now add a list of more recent references. See Refs. 15,18,19,20

Round 2

Reviewer 2 Report

The authors corrected the reviewer's criticism in the second version of the manuscript. As a result, the manuscript now shows more interest for the reader and offers the quality of the presentation.